# Primary care clinical management following self-harm during the first wave of COVID-19 in the UK: population-based cohort study

Sarah Steeg [ID],[1,2,3] Matthew Carr,[1,2,4,5] Laszlo Trefan,[1,2] Darren Ashcroft,[2,3,4,5] Navneet Kapur,[1,2,5,6] Emma Nielsen,[7] Brian McMillan [ID],[8] Roger Webb[1,2,5]

SS and MC are joint first authors.

For numbered affiliations see end of article.

**Correspondence to**
Dr Sarah Steeg;
sarah.steeg@manchester.ac.uk

## ABSTRACT

**Objectives** A substantial reduction in self-harm recorded in primary care occurred during the first wave of COVID-19 but effects on primary care management of self-harm are unknown. Our objectives were to examine the impact of COVID-19 on clinical management within 3 months of an episode of self-harm.

**Design** Retrospective cohort study.

**Setting** UK primary care.

**Participants** 4238 patients with an index episode of self-harm recorded in UK primary care during the COVID-19 first-wave period (10 March 2020–10 June 2020) compared with 48 739 patients in a prepandemic comparison period (10 March–10 June, 2010–2019).

**Outcome measures** Using data from the UK Clinical Practice Research Datalink, we compared cohorts of patients with an index self-harm episode recorded during the prepandemic period versus the COVID-19 first-wave period. Patients were followed up for 3 months to capture subsequent general practitioner (GP)/practice nurse consultation, referral to mental health services and psychotropic medication prescribing. We examined differences by gender, age group and Index of Multiple Deprivation quintile.

**Results** Likelihood of having at least one GP/practice nurse consultation was broadly similar (83.2% vs 80.3% in the COVID-19 cohort). The proportion of patients referred to mental health services in the COVID-19 cohort (4.2%) was around two-thirds of that in the prepandemic cohort (6.1%). Similar proportions were prescribed psychotropic medication within 3 months in the prepandemic (54.0%) and COVID-19 first-wave (54.9%) cohorts.

**Conclusions** Despite the challenges experienced by primary healthcare teams during the initial COVID-19 wave, prescribing and consultation patterns following self-harm were broadly similar to prepandemic levels. We found no evidence of widening of digital exclusion in terms of access to remote consultations. However, the reduced likelihood of referral to mental health services warrants attention. Accessible outpatient and community services for people who have self-harmed are required as the COVID-19 crisis recedes and the population faces new challenges to mental health.

## INTRODUCTION

Prompt clinical intervention and follow-up is recommended for people who have recently

### Strengths and limitations of this study

► First study examining effects of COVID-19 on primary care management following self-harm in the UK.
► Findings are based on a large number of general practices across the UK, using data from the Clinical Practice Research Datalink.
► Data broadly representative of geographical coverage, area-level deprivation, age and sex in England.
► Data accuracy is determined by the quality of the information inputted by contributing practices.

self-harmed, in part due to their increased risks of suicide.[1] Self-harm includes intentional self-poisoning and self-injury and can involve varying degrees of suicidal intent.[2] All episodes of self-harm should be followed by comprehensive mental health assessment to identify psychosocial needs and address risks of further self-harm and suicide.[1] This has become more challenging due to the disruption to UK health services caused by the COVID-19 pandemic. Fluctuations in infection rates, lockdown restrictions and public health messaging has had a significant impact on the numbers of people accessing primary care services, with considerable reductions in presentation rates found for a number of physical and mental health conditions,[3] including self-harm.[4] The pandemic and its pervasive impacts on everyday life has also had a detrimental impact on the mental health of the population[5] and the first wave of COVID-19 and lockdown may have led to an increase in the prevalence of suicidal ideation.[6] This unique combination of factors has led to a shortfall in the numbers of people receiving support from healthcare services after harming themselves.

In April 2020, during the first UK COVID-19 lockdown, rates of primary care-recorded incident self-harm in the UK were 38% lower

than expected based on trends that occurred during the previous 10 years.[4] While rates of help seeking gradually returned towards expected levels through the subsequent months up to September 2020, further regional COVID-19 containment restrictions and national lockdowns from autumn 2020 into winter 2021 are likely to have additionally affected rates of help seeking. Some people sought help from alternative sources while the UK was in its first lockdown; for example, some mental health charities reported increases in demand for services such as helplines.[7] One study in the US reported emergency hospital presentations following suicide attempts had increased to higher than prepandemic levels following an initial reduction,[8] suggesting that observed reductions in help seeking in primary care did not necessarily reflect population need. Furthermore, a living systematic review on the impacts of COVID-19 on suicidal behaviour globally found that while the majority of studies reported a decrease in health service contacts for self-harm, some identified an increased likelihood of using more lethal methods.[9]

Primary care settings provide vital support for people who have self-harmed. A recent study found that 26% of people sought help from their general practitioner (GP) in the week prior to suicide, with self-harm a common reason for contact.[10] Previous research found that 15% of patients with an episode of self-harm recorded in primary care were referred to mental health services from their GP within a year.[11] A recent report found that, among people referred to mental health services, their GP was the most common referral route.[12] However, this report highlighted many barriers that exist for people accessing support following self-harm.[12] GPs, as well as patients, have reported struggling to find appropriate self-harm services, with limited referral options and shortages in community services identified.[13]

It is unknown how clinical management following self-harm has been affected by the COVID-19 pandemic. Although GPs continued to offer consultations to patients throughout the first wave in the spring of 2020, with most consultations taking place remotely,[14] recently conducted research found lower rates of GP referral to mental health services following primary care-recorded common mental illnesses and self-harm episodes.[4] There has also been concern that the pandemic could contribute further to digital exclusion; patients living with greater levels of socioeconomic deprivation being further excluded from remote clinical care.[15] The degree to which primary care management of people who had self-harmed was impacted during the first COVID-19 wave is unknown. This is an important research question, because self-harm is a key risk factor for suicide and requires specific timely intervention.[16]

In the UK, a nationwide lockdown was imposed on 23 March 2020, with public health messaging to encourage people to avoid contact with others announced the week before this.[17] We aimed to examine clinical management of self-harm during the 3 months following the beginning of the UK COVID-19 containment measures and national lockdown, using data from a prepandemic comparison period to examine effects of COVID-19. Our specific objectives were to:

1. Identify two cohorts of patients presenting with an index episode of self-harm, comprising prepandemic versus COVID-19 first-wave time periods.
2. Estimate the probabilities (%) of receiving a new psychotropic medication prescription (by drug type), frequency of subsequent GP or practice nurse consultations, and referral to mental health services within 3 months of the index self-harm episode.
3. Estimate ratios comparing probabilities of psychotropic medication prescribing, referral to mental health services and GP or practice nurse consultation (by face to face and telephone) within 3 months of the index episode, between the COVID-19 and prepandemic comparison cohorts.

## METHODS
### Study design, data sources, and participants
We conducted a cohort study using anonymised primary care data from the Clinical Practice Research Datalink (CPRD) Aurum and GOLD databases.[18 19] Both Aurum and GOLD contain data extracted from electronic patient record platforms; specifically, EMIS and Vision. These platforms are used to record information about patients including content of patient consultations, signs and symptoms, diagnoses, tests, medication prescriptions and referrals. The Aurum database includes general practices based in England that contribute data using the EMIS clinical system. The GOLD database is extracted from the Vision system, with most of its contributing general practices based in Northern Ireland, Scotland and Wales. The Aurum database is broadly representative of geographical coverage, area-level deprivation, age and sex distributions of the population of England,[18] whist GOLD is broadly representative of the sociodemographic profile of the whole UK population.[19] To avoid including duplicate general practices in the Aurum and GOLD databases, we excluded English practices in our analyses of GOLD data. The CPRD includes information on patient demographics, consultations, symptoms, diagnoses, medication prescriptions and referrals to secondary care. We also obtained information on Index of Multiple Deprivation (IMD) score linked at general practice level.[20] The IMD is a single score derived from seven domains of areas-based deprivation measures: income, employment, education, health, crime, barriers to housing and services, and living environment.[21] The IMD is a relative measure of deprivation between areas. The postcode of the general practice was linked to the IMD score of its corresponding lower super output area in England and Wales (an area typically containing around 1500 residents), super output area in Northern Ireland (containing an average of 2100 individuals) or datazone in Scotland (containing a population of between 500 and 1000).

**Table 1** Characteristics of cohorts with a primary care-recorded episode of self-harm in the UK

| | Comparison cohort (2010–2019) | COVID-19 cohort (2020) |
|---|---|---|
| Total index episodes (between 10 March and 10 June) | 48739 | 4238 |
| Gender: | | |
| Female | 29596 (60.7%) | 2510 (59.2%) |
| Male | 19143 (39.3%) | 1728 (40.8%) |
| Age group (years): | | |
| 10–24 | 20308 (41.7%) | 1844 (43.5%) |
| 25–64 | 25700 (52.7%) | 2159 (50.9%) |
| ≥65 | 2731 (5.6%) | 235 (5.6%) |
| Practice-level IMD quintile: | | |
| 1 (least deprived) | 5687 (11.7%) | 510 (12.0%) |
| 2 | 7003 (14.4%) | 600 (14.2%) |
| 3 | 8035 (16.5%) | 729 (17.2%) |
| 4 | 10837 (22.2%) | 980 (23.1%) |
| 5 (most deprived) | 12678 (26.0%) | 981 (23.2%) |
| Unknown | 4499 (9.2%) | 438 (10.3%) |

IMD, Index of Multiple Deprivation.

Analysis was conducted on pooled Aurum and GOLD data. We compared two cohorts of patients: (1) those with an index self-harm episode recorded in a prepandemic comparison period (between 10 March to 10 June, 2010-2019) and those with an index self-harm episode recorded in the COVID-19 first-wave period (10 March 2020–10 June 2020). Patients in each cohort were followed up for 3 months to capture psychotropic medication prescribing, referral to mental health services and GP or practice nurse consultations. To be included in either of the two study cohorts, patients must have been aged 10 years or older and registered with a contributing practice, deemed by the CPRD as providing up-to-standard data, for at least 1 year prior to the date of the index self-harm episode. Patients with less than 3 months of follow-up time in the CPRD were excluded from our analyses. The cohorts were restricted to patients with records that were deemed acceptable by the CPRD for research purposes, which excluded patients with missing data on sex or age. IMD data were missing for 9.2% of the prepandemic comparison cohort and 10.3% of the COVID-19 first-wave cohort (table 1). In terms of missing outcome data, if there was no record of psychotropic medication prescribing, face-to-face or remote consultation and referral to mental health service, the outcome was not recorded as present.

## Exposures, outcomes and covariates

Index episodes of self-harm were identified from Read, SNOMED and EMIS codes: SNOMED CT[22] is a clinical vocabulary that is readable by computers. Used internationally, it is the recommended structured clinical vocabulary to record electronic patient information in the National Health Service (NHS) in England. EMIS and Read codes are further coding systems used to capture clinical terms used in patient records.[23] Codes relating to intentional self-poisoning and self-injury episodes, of varying degrees of suicidal intent, were included. To identify individuals' first record of self-harm, we applied a retrospective analysis period during which a patient was required to have been registered at the practice for at least 1 year before an incident episode. We examined any psychotropic medication and specific psychotropic medication types including antidepressants, antipsychotics, anxiolytic/hypnotics, mood stabilisers and stimulants (https://clinicalcodes.rss.mhs.man.ac.uk/medcodes/article/173/). Mode of consultation was grouped into face-to-face and video/telephone consultations, with a GP or practice nurse. Information on referral to mental health services were identified using two CPRD fields: a 'psychiatry' code in the Family Health Services Authority (FHSA) specialty variable and codes of 'mental illness', 'child and adolescent psychiatry', 'forensic psychiatry', 'psychotherapy', 'old age psychiatry', 'clinical psychology', 'adult psychiatry' and 'community psychiatric nurse' in the NHS specialty field.[11] We combined information from both the FHSA and NHS specialty fields to construct a binary specialist mental health services referral indicator. All code lists were verified by senior clinical academics as part of a previous study[4] and are available online (https://clinicalcodes.rss.mhs.man.ac.uk/medcodes/article/173/). We examined frequencies and rate ratios between the prepandemic and COVID-19 first-wave cohorts by the following covariates: gender, age group (10–24, 25–64 and 65 and older) and practice-level IMD quintile. Subgroup categories were derived to avoid reporting cell counts less than 10; if cell counts were found to be less than 10, subgroups were collapsed.

## Analysis

Frequencies and probabilities of GP/nurse consultation, referral to mental health services and psychotropic medication prescribing during the antecedent period were estimated and compared with observed values during the COVID-19 first-wave period. The modelling was conducted using modified Poisson regression in a generalised linear modelling framework with a log-link function and a robust variance estimator with analyses stratified by gender, age group and practice-level IMD quintile. This study was conducted in accordance with REporting of studies Conducted using Observational Routinely collected health Data guidance ((online supplemental table 1).[24]

## Patient and public involvement

A panel of four service users and carers with lived experience of health services following self-harm collaborated with the research team to plan the study and interpret results. Panel members reviewed findings based on their experiences of health services for self-harm and the COVID-19 pandemic

and associated societal restrictions. Over two workshops, panel members met with the corresponding author of the study to review the results of the study and provide feedback on their visual presentation. The group is linked with the National Institute Health Research Greater Manchester Patient Safety Translational Research Centre.

## RESULTS

A total of 48 739 patients had an index episode of self-harm recorded in the UK during the prepandemic comparison period and 4238 were recorded in the COVID-19 first-wave cohort (table 1). The gender, age and deprivation profiles of the two cohorts were broadly similar, with the majority of recorded self-harm episodes by women and more self-harm episodes recorded in practices in areas of higher deprivation.

Unsurprisingly, the likelihood of patients receiving a remote GP/practice nurse consultation within 3 months of a self-harm episode was higher in the COVID-19 'first-wave' cohort (67.7%) than in the prepandemic comparison cohort (32.3%, ratio 2.10, 95% CI 2.05 to 2.15) (table 2). Although the overall likelihood of having a GP/practice nurse consultation was slightly lower in the COVID-19 cohort (80.3%) than in the prepandemic comparison cohort (83.2%), ratio 0.97, 95% CI 0.96 to 0.98, this pattern did not apply to all demographic groups. Men, patients aged 65 years and over and those registered with practices in the two most deprived quintiles were equally likely to have had a GP/practice nurse consultation in the prepandemic and COVID-19 first-wave cohorts. With respect to remote consultation specifically, there was no difference between practice deprivation-level and likelihood of receiving this form of management.

Overall, 4.2% of patients (179/4238) were referred to mental health services in the COVID-19 first-wave cohort, a significant reduction vs the probability observed in the prepandemic comparison cohort (6.1%, ratio 0.70, 95% CI 0.60 to 0.81) (table 3). The reduction in likelihood of being referred to mental health servicers was not observed for patients aged 65 years and over (ratio 1.66, 95% CI 0.98 to 2.80, p value for effect modification by age group=0.01).

Just over half of patients in the prepandemic (54.0%) and COVID-19 first-wave (54.9%) cohorts were prescribed psychotropic medication within 3 months of the index self-harm episode (table 4). The likelihoods of receiving such treatment were broadly similar across gender and deprivation quintiles. Among patients aged 10–24 years, those in the COVID-19 first-wave cohort were more likely to be prescribed psychotropic medication (ratio 1.14, 95% CI 1.07 to 1.22). Considering prescriptions for antidepressant medication specifically, probabilities were broadly similar in the prepandemic and COVID-19 first-wave cohorts, though they were higher among young people aged 10–24 years in the COVID-19 first-wave cohort (ratio 1.18, 95% CI 1.10 to 1.26). Details on prescribing of antipsychotic, anxiolytic/hypnotic, mood stabilisers and stimulants are in table 5.

## DISCUSSION
### Summary

Prepandemic and COVID-19 first-wave cohorts of patients with an index episode of self-harm recorded in primary care had similar gender, age and deprivation profiles. Similar proportions were prescribed psychotropic medication within 3 months of their index self-harm episode—just over half in both the prepandemic and COVID-19 first-wave cohorts. However, patients aged 10–24 years in the COVID-19 first-wave cohort were more likely to be prescribed psychotropic medication than in the preceding years. Overall, the likelihood of having at least one GP/practice nurse consultation was slightly lower in the COVID-19 first-wave cohort, although there was no such difference observed among men, patients aged 65 years and over, and those registered with practices located in more deprived areas. Patients in more deprived practice populations who had harmed themselves were just as likely to consult face to face or remotely with a GP or a practice nurse than those in less deprived populations. The proportion of patients referred to mental health services in the COVID-19 first-wave cohort was around two-thirds of that in the prepandemic comparison cohort.

### Strengths and limitations

The main strength of our study is the broadly representative data source that included a large number of general practices across the UK. CPRD Aurum is broadly representative of geographical coverage, area-level deprivation, age and sex in England[18] while CPRD GOLD dataset is broadly representative of the UK age and sex profile.[19] This enables us to make inferences at national level about how the pandemic affected primary care clinical management of patients who have self-harmed. However, our findings may not be generalisable to countries experiencing different degrees of COVID-19 containment measures and societal restrictions and those with much lower levels of access to universal healthcare. There are some limitations in utilising anonymised primary care records. The data are extracted from GP information systems and their accuracy is determined by the quality of the information inputted by contributing practices. The rapid adaptations to working methods that were necessary during the early stages of the pandemic may have affected accuracy of clinical coding. Some of the self-harm episodes recorded in primary care would have been emergency department presentations that were subsequently added to the patient's primary care record. Suicidal intent specific to each self-harm episode could not be examined in this study. We were also unable to examine clinical management outcomes with no corresponding referral code, such as patients being advised to self-refer to third sector organisations or Improving Access to Psychological Therapies services. Future research using linked Hospital Episode Statistics will enable separate examination of emergency department self-harm presentations. Finally, we were unable to examine suicide deaths and other causes of mortality in this study due to unavailability of linked mortality records at the time of analysis.

**Table 2** GP/practice nurse consultations following a primary care-recorded episode of self-harm in the UK

| | Face-to-face consultation | | | Telephone/videoconsultation | | | Face-to-face or telephone/videoconsultation | | |
|---|---|---|---|---|---|---|---|---|---|
| | Comparison cohort, % (n) | COVID-19 cohort, % (n) | Ratio (95% CI) | Comparison cohort, % (n) | COVID-19 cohort, % (n) | Ratio (95% CI) | Comparison cohort, % (n) | COVID-19 cohort, % (n) | Ratio (95% CI) |
| All persons | 80.1 (39 027) | 54.3 (2300) | 0.68 (0.66 to 0.70) | 32.3 (15 723) | 67.6 (2866) | 2.10 (2.05 to 2.15) | 83.2 (40 536) | 80.3 (3403) | 0.97 (0.95 to 0.98) |
| Gender: | | | | | | | | | |
| Female | 82.5 (24 423) | 56.3 (1412) | 0.68 (0.66 to 0.71) | 33.8 (10 004) | 69.8 (1752) | 2.06 (2.00 to 2.13) | 85.5 (25 300) | 82.1 (2060) | 0.96 (0.94 to 0.98) |
| Male | 76.3 (14 604) | 51.4 (888) | 0.67 (0.64 to 0.71) | 29.9 (5719) | 64.5 (1114) | 2.16 (2.07 to 2.25) | 79.6 (15 236) | 77.7 (1343) | 0.98 (0.95 to 1.00) |
| Age group (years): | | | | | | | | | |
| 10–24 | 74.3 (15 083) | 48.2 (888) | 0.65 (0.62 to 0.68) | 24.9 (5055) | 60.5 (1116) | 2.43 (2.33 to 2.54) | 77.4 (15 720) | 74.7 (1377) | 0.96 (0.94 to 0.99) |
| 25–64 | 84.0 (21 596) | 58.0 (1252) | 0.69 (0.67 to 0.72) | 36.4 (9341) | 72.8 (1571) | 2.00 (1.94 to 2.06) | 86.9 (22 327) | 84.0 (1813) | 0.97 (0.95 to 0.99) |
| ≥65 | 86.0 (2,348) | 68.1 (160) | 0.79 (0.72 to 0.87) | 48.6 (1327) | 76.2 (179) | 1.57 (1.45 to 1.70) | 91.1 (2,489) | 90.6 (213) | 0.99 (0.95 to 1.04) |
| Practice-level IMD quintile: | | | | | | | | | |
| 1 (least deprived) | 82.4 (4687) | 56.9 (290) | 0.69 (0.64 to 0.74) | 37.5 (2135) | 68.0 (347) | 1.81 (1.69 to 1.94) | 86.0 (4891) | 81.0 (413) | 0.94 (0.90 to 0.98) |
| 2 | 81.6 (5711) | 50.8 (305) | 0.62 (0.58 to 0.67) | 33.4 (2340) | 70.2 (421) | 2.10 (1.97 to 2.23) | 84.5 (5918) | 80.0 (480) | 0.95 (0.91 to 0.99) |
| 3 | 81.3 (6534) | 55.8 (407) | 0.69 (0.64 to 0.73) | 34.0 (2731) | 66.3 (483) | 1.95 (1.84 to 2.07) | 84.7 (6807) | 80.0 (583) | 0.94 (0.91 to 0.98) |
| 4 | 79.5 (8613) | 53.7 (526) | 0.68 (0.64 to 0.72) | 32.0 (3472) | 67.5 (661) | 2.11 (2.00 to 2.22) | 82.6 (8949) | 80.8 (792) | 0.98 (0.95 to 1.01) |
| 5 (most deprived) | 79.1 (10 023) | 56.2 (551) | 0.71 (0.67 to 0.75) | 29.3 (3717) | 68.3 (670) | 2.33 (2.21 to 2.45) | 81.7 (10 358) | 81.9 (803) | 1.00 (0.97 to 1.03) |
| Unknown | 76.9 (3459) | 50.5 (221) | 0.66 (0.60 to 0.72) | 29.5 (1328) | 64.8 (284) | 2.20 (2.02 to 2.39) | 80.3 (3613) | 75.8 (332) | 0.94 (0.89 to 1.00) |

IMD, Index of Multiple Deprivation.

**Table 3** Referrals to mental health services following a primary care-recorded episode of self-harm in the UK

| | Comparison cohort: % (n/N) | COVID-19 cohort: % (n/N) | Ratio (95% CI) |
|---|---|---|---|
| All persons | 6.1 (2959/48 739) | 4.2 (179/4238) | 0.70 (0.60 to 0.81) |
| Gender: | | | |
| Female | 6.4 (1888/29 596) | 3.9 (99/2510) | 0.62 (0.51 to 0.75) |
| Male | 5.6 (1071/19 143) | 4.6 (80/1728) | 0.83 (0.66 to 1.03) |
| Age group (years): | | | |
| 10–24 | 6.7 (1364/20 308) | 4.2 (77/1844) | 0.62 (0.50 to 0.78) |
| 25–64 | 5.8 (1490/25 700) | 4.0 (87/2159) | 0.70 (0.56 to 0.86) |
| ≥65 | 3.8 (105/2731) | 6.4 (15/235) | 1.66 (0.98 to 2.80) |
| Practice-level IMD quintile: | | | |
| 1 (lowest) | 7.4 (420/5687) | 6.3 (32/510) | 0.85 (0.60 to 1.20) |
| 2 | 6.8 (475/7003) | 4.3 (26/600) | 0.64 (0.43 to 0.94) |
| 3 | 6.4 (512/8035) | 4.1 (30/729) | 0.65 (0.45 to 0.93) |
| 4 | 5.8 (629/10 837) | 3.5 (34/980) | 0.60 (0.43 to 0.84) |
| 5 (highest) | 4.9 (623/12 678) | 3.8 (37/981) | 0.77 (0.55 to 1.06) |

IMD, Index of Multiple Deprivation.

## Comparison with existing literature

Previous research found that reductions in help-seeking were greatest among patients registered at practices located in more deprived areas.[4] Our study found that patients at practices in the two most deprived quintiles who did seek help had similar rates of psychotropic medication prescribing as those at practices in less deprived areas, in both the prepandemic comparison and COVID-19 first-wave cohorts. Similarly, while there was a clear deprivation gradient in likelihood of remote consultation for self-harm prior to the pandemic, with those in areas of lower deprivation less likely to have remote consultation, the abrupt switch to remote consultations once the COVID-19 crisis had commenced did not lead to widening of existing inequalities in this respect. Evidence

**Table 4** Prescriptions of any type of psychotropic medication and any antidepressant drug following a primary care-recorded episode of self-harm in the UK

| | Any psychotropic medication prescription | | | Any antidepressant prescription | | |
|---|---|---|---|---|---|---|
| | Comparison Cohort, % (n) | COVID-19 cohort, % (n) | Ratio (95% CI) | Comparison cohort, % (n) | COVID-19 cohort, % (n) | Ratio (95% CI) |
| All persons | 54.0 (26 317) | 54.9 (2328) | 1.02 (0.99 to 1.05) | 47.7 (23 243) | 49.4 | 1.04 (1.00 to 1.07) |
| Gender: | | | | | | |
| Female | 53.6 (15 850) | 53.9 (1354) | 1.01 (0.97 to 1.05) | 48.5 (14 338) | 49.4 | 1.02 (0.98 to 1.06) |
| Male | 54.7 (10 467) | 56.4 (974) | 1.03 (0.99 to 1.08) | 46.5 (8905) | 49.4 | 1.06 (1.01 to 1.12) |
| Age group (years): | | | | | | |
| 10–24 | 30.9 (6265) | 35.1 (648) | 1.14 (1.07 to 1.22) | 26.7 (5430) | 31.5 | 1.18 (1.10 to 1.26) |
| 25–64 | 70.8 (18 195) | 70.2 (1515) | 0.99 (0.96 to 1.02) | 63.3 (16 274) | 63.8 | 1.01 (0.97 to 1.04) |
| ≥65 | 68.0 (1857) | 70.2 (165) | 1.03 (0.95 to 1.13) | 56.4 (1539) | 58.7 | 1.04 (0.93 to 1.17) |
| Practice-level IMD quintile: | | | | | | |
| 1 (least deprived) | 54.1 (3078) | 53.7 (274) | 0.99 (0.91 to 1.08) | 48.4 (2752) | 49.2 (251) | 1.02 (0.93 to 1.12) |
| 2 | 54.0 (3789) | 55.0 (330) | 1.02 (0.94 to 1.10) | 47.9 (3352) | 49.5 (297) | 1.03 (0.95 to 1.13) |
| 3 | 54.5 (4378) | 56.8 (414) | 1.04 (0.98 to 1.11) | 48.3 (3883) | 50.3 (367) | 1.04 (0.97 to 1.12) |
| 4 | 54.2 (5872) | 55.0 (539) | 1.02 (0.96 to 1.08) | 47.5 (5152) | 49.8 (488) | 1.05 (0.98 to 1.12) |
| 5 (most deprived) | 53.3 (6758) | 55.8 (547) | 1.05 (0.99 to 1.11) | 46.9 (5945) | 50.0 (490) | 1.07 (1.00 to 1.14) |
| Unknown | 54.3 (2442) | 51.1 (224) | 0.94 (0.86 to 1.04) | 48.0 (2159) | 46.1 (202) | 0.96 (0.86 to 1.07) |

IMD, Index of Multiple Deprivation.

**Table 5** Prescriptions of specific psychotropic medication types following a primary care-recorded episode of self-harm in the UK

| | Comparison cohort, % (n) | COVID-19 cohort, % (n) | Ratio (95% CI) |
|---|---|---|---|
| Antidepressants | 47.7 (23 243) | 49.4 (2095) | 1.04 (1.00 to 1.07) |
| Antipsychotics | 10.8 (5285) | 12.3 (521) | 1.13 (1.04 to 1.23) |
| Anxiolytics/hypnotics | 17.6 (8577) | 15.5 (657) | 0.88 (0.82 to 0.95) |
| Mood stabilisers | 5.7 (2763) | 5.6 (238) | 0.99 (0.87 to 1.13) |
| Stimulants | 0.8 (369) | 1.0 (44) | 1.37 (1.00 to 1.87) |

conducted prior to the COVID-19 pandemic indicated there was potential for the growth in virtual consultations to widen existing disparities in access to health.[25] Our findings suggest that for people who did seek help, there was no exacerbation of existing inequalities due to COVID-19 in these particular aspects of primary care clinical management, despite the growth in remote consultation numbers. In other words, we found no evidence of digital exclusion from self-harm care during the pandemic due to socioeconomic influences.

Likelihood of referral to NHS mental health services was lower during the first wave of COVID-19 than during the prepandemic comparison period, particularly for patients aged under 65 years. Younger people have been found to have been particularly negatively affected by the pandemic. Individuals aged 18–29 years reported greater increases in suicidal ideation over the first 6 weeks of the UK's lockdown than other groups,[6] and were found to have the greatest deteriorations in mental health.[5] Furthermore, working age adults were previously identified as having the greatest reductions in help seeking for mental illness and self-harm during April 2020.[4]

### Implications for research and practice

Evidence shows the number of people seeking mental health help, including for self-harm, from non-NHS services such as via digital platforms and helplines increased during the second quarter of 2020.[26 27] This has implications for the clinical guidance for people who have self-harmed, which recommends that assessment by a mental health specialist should follow all episodes of self-harm.[16] Furthermore, such assessments might be particularly challenging in non-face-to-face settings.

Data from NHS Digital[28] showed a 10% decrease in the number of new referrals to NHS mental health services in the 6 months from 1 April 2020. During the same time period, there was an increase in antidepressant prescribing of around 4% leading to concerns that the increased mental health burden caused by the COVID-19 crisis could be being managed pharmacologically rather than with psychosocial interventions. Our findings suggest this could also have been happening with patients who have harmed themselves. While outpatient and community mental health services adapted to provide alternatives to face-to-face support, our findings suggest that primary care practitioners were less inclined to refer to these services during the early stages of the pandemic. Ongoing work in England to improve community support following self-harm emphasises a need

to better align the voluntary sector with primary healthcare services,[29] to ensure that GPs are equipped with the information that they require to make referrals to community self-harm resources as well as NHS services.

### Conclusions

Despite the challenges experienced by GPs in delivering healthcare during the 3 months of the initial wave of COVID-19 in the UK, the management of self-harm was broadly similar to the prepandemic comparison period in terms of psychotropic prescribing and GP/nurse consultations. We found no evidence of digital exclusion due to socioeconomic influences during the pandemic, in terms of likelihood of remote consultation. However, the reduced likelihood of referral to mental health services during March–June 2020 warrants close attention, particularly in the context of increasing prevalence of mental distress in the population. Our findings suggest that COVID-19 may have increased the likelihood that adolescents and young people do not receive psychosocial interventions, with pharmacological intervention alone becoming more likely during the early phase of the crisis. Accessible outpatient and community services that people who have self-harmed can be referred to by their GPs are required.

**Author affiliations**
[1]Centre for Mental Health and Safety, University of Manchester, Manchester, UK
[2]Manchester Academic Health Science Centre, University of Manchester, Manchester, UK
[3]National Institute for Health Research School for Primary Care Research, University of Manchester, Manchester, UK
[4]Centre for Pharmacoepidemiology and Drug Safety, University of Manchester, Manchester, UK
[5]National Institute for Health Research Greater Manchester Patient Safety Translational Research Centre, University of Manchester, Manchester, UK
[6]Greater Manchester Mental Health NHS Foundation Trust, Manchester, UK
[7]Self-harm Research Group, School of Psychology, University of Nottingham, Nottingham, UK
[8]Centre for Primary Care and Health Services Research, University of Manchester, Manchester, UK

**Acknowledgements** We thank Stephen Barlow, Elizabeth Monaghan, Fiona Naylor, and Jonathan Smith (members of the Centre for Mental Health and Safety patient and public involvement and engagement group of the NIHR Greater Manchester Patient Safety Translational Research Centre) for their advisory roles in the study. We would also like to acknowledge all the data providers and general practices that made the anonymised data available for research.

**Contributors** SS, MC, LT, DA, NK, EN, BM and RW conceptualised the study and contributed to its design. MC, LT and SS acquired, accessed and verified the data.

MC generated the clinical code lists. MC and LT managed the data and conducted the statistical analysis. SS, MC, LT, DA, NK, EN, BM and RW interpreted the results. SS drafted the manuscript. SS, MC, LT, DA, NK, EN, BM and RTW critically reviewed the manuscript and approved the final version. SS accepts full responsibility for the finished work and the conduct of the study, had access to the data and controlled the decision to publish.

**Funding** This work was funded by a UK Research and Innovation COVID-19 Rapid Response Initiative grant (grant reference COV0499), a University of Manchester Presidential Fellowship (awarded to SS) and the NIHR Greater Manchester Patient Safety Translational Research Centre.

**Competing interests** NK reports grants and personal fees from the UK Department of Health and Social Care, the National Institute of Health Research (NIHR), the National Institute for Health and Care Excellence (NICE) and the Healthcare Quality and Improvement Partnership, outside the submitted work; works with NHS England on national quality improvement initiatives for suicide and self-harm; is a member of the advisory group for the National Suicide Prevention Strategy of the Department of Health and Social Care; has chaired NICE guideline committees for Self-harm and Depression; and is currently the Topic Advisor for the new NICE Guidelines for self-harm. All other authors declare no competing interests.

**Patient and public involvement** Patients and/or the public were involved in the design, or conduct, or reporting, or dissemination plans of this research. Refer to the Methods section for further details.

**Patient consent for publication** Not applicable.

**Ethics approval** The study was approved by the Independent Scientific Advisory Committee (protocol number 20_001RA2).

**Provenance and peer review** Not commissioned; externally peer reviewed.

**Data availability statement** No data are available. The clinical codes used in this study are available online. The codes are also available from the corresponding author on request. Access to dataare available only once approval has been obtained through the individual constituent entities controlling access to the data. The primary care data can be requested via application to the Clinical Practice Research Datalink.

**ORCID iDs**
Sarah Steeg http://orcid.org/0000-0002-7935-1414
Brian McMillan http://orcid.org/0000-0002-0683-3877

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
