## [Reviewer comments · BMJ Open]

ARTICLE DETAILS

TITLE (PROVISIONAL)	Primary care clinical management following self-harm during the first wave of COVID-19 in the UK: population-based cohort study
AUTHORS	Steeg, Sarah; Carr, M; Trefan, Laszlo; Ashcroft, Darren; Kapur, Navneet; Nielsen, Emma; McMillan, Brian; Webb, Roger

VERSION 1 – REVIEW

REVIEWER	Calati, Raffaella University of Milan–Bicocca
REVIEW RETURNED	13-Sep-2021

GENERAL COMMENTS	The present study examined the impact of COVID-19 on clinical management within three months of an episode of self-harm. Authors compared 4,238 patients with an index episode of self-harm recorded in UK primary care during the COVID-19 first-wave period versus 48,739 patients in a pre-pandemic comparison period. The aim is timely and extremely interesting. However, the writing is not always clear, the results should be better discussed in the light of possible biases and put in perspectives accordingly to data of the entire world and not only referring to the UK situation. Authors may want to follow these suggestions to improve their manuscript. Abstract Minor remark: line 1: “substantial self-harm reduction” instead of “substantial reduction self-harm” or am I wrong? Introduction The introduction is very well written. Methods I am not sure the writing “a prospective cohort study” is correct since in the abstract this study is described as retrospective. The dates of the beginning and the ending of the analyses would help to clarify. EMIS® and Vision® systems should be described for readers outside UK. Authors wrote: “Patients with less than 3 months of follow-up time in the CPRD were excluded from our analyses”. This point should be discussed as a potential bias because most severe patients could have been lost. Exposures, outcomes and covariates should be described in detail. Read, SNOMED and EMIS codes should be briefly explained. The same for FHSA and NHS fields.
--

	Authors wrote: “A panel of four service users and carers with lived experience of health services following self-harm collaborated with the research team to plan the study and interpret results”. May you explain a little bit more? Thanks. Please add power analysis. IMD should be better described. What about missing data in the tables? Authors mentioned only sex, age and IMD. Discussion Authors should discuss results trying to consider data of other countries/continents.
--	--

REVIEWER	Younès, Nadia EA 40-47 University of Versailles Saint-Quentin
REVIEW RETURNED	06-Oct-2021

GENERAL COMMENTS	The manuscript presents an interesting and well-described population-based cohort study in primary care about clinical management following self-harm during the first wave of Covid-19 in UK. Results are important to describe that clinical management and referral to mental health service in that particular period. It is interesting to note that the management of self-harm was broadly similar to the management in the pre-pandemic period, with less referral to mental health services. Few remarks can be formulated.  - The management of self-harm includes a comprehensive mental assessment but also a management of the clinical situation associated with self-harm. I am embarrassed that care seems to be limited to the prescription of psychotropic drugs. GPs consultations and practice nurse consultations are the main component. I suggest changing the order of the objectives (and in the presentation of results, in tables) and putting (iii) as the second objective and (ii) as the third.... - Is it possible to know how many GPs consultations and practice nurse consultations was realized? - Is it possible to present data about the other diagnosis (than self-harm) established by GPs? About suicidal intent? - Results about psychotropic medication type (almost antidepressant) should be presented not only in supplementary material. They deserve to be presented in the abstract, the results and in the discussion (comparison with others studies). - It would be interesting to describe further the four service users and carers' involvement in the study
---

VERSION 1 – AUTHOR RESPONSE

Reviewer 1			
3	The present study examined the impact of COVID-19 on clinical management within three months of an episode of self-harm. Authors	We have addressed both of these concerns in our response. We have added to our paper to ensure there is a global perspective: 'Strengths and limitations':	

	compared 4,238 patients with an index episode of self-harm recorded in UK primary care during the COVID-19 first-wave period versus 48,739 patients in a pre-pandemic comparison period. The aim is timely and extremely interesting. However, the writing is not always clear, the results should be better discussed in the light of possible biases and put in perspectives accordingly to data of the entire world and not only referring to the UK situation.	“Finally, our findings may not be generalisable to countries experiencing different forms and degrees of COVID-19 restriction measures, and those with much lower levels of access to universal healthcare.” We have also added discussion of changes in health service use for self-harm globally to the Introduction: “Furthermore, a living systematic review on the impacts of COVID-19 on suicidal behaviour globally found that, although the majority of studies reported a decrease in health service contacts for self-harm, some identified an increase in more lethal methods.” In terms of further discussion of possible biases, we have added the following to the Discussion “The rapid adaptations to working methods that were necessary during the early stages of the pandemic may have affected accuracy of clinical coding of consultations and referrals.” However, our findings do not suggest that this was the case. Other sources of bias, such as selection and non-participation biases are not relevant to CPRD-based studies and we are confident that we have addressed potential sources of bias.	10 4 9
4	Abstract Minor remark: line 1: “substantial self-harm reduction” instead of “substantial reduction self-harm” or am I wrong?	Thank you for noting this. There was a missing word in this sentence; it has been amended to say: “A substantial reduction in the frequency of self-harm episodes recorded in primary care...”	Abstract, page 2
5	Introduction The introduction is very well written.	Thank you for this positive feedback.	

6	Methods I am not sure the writing “a prospective cohort study” is correct since in the abstract this study is described as retrospective. The dates of the beginning and the ending of the analyses would help to clarify.	Thank you for noting this error – we now simply refer to the study as a ‘cohort study’, as described in the Abstract. We have corrected the term ‘prospective’ in the Methods section.	Methods, ‘Study design, data sources, and participants’, page 6.
7	EMIS® and Vision® systems should be described for readers outside UK.	We have added a description of EMIS® and Vision®: “Both Aurum and GOLD databases are based on data extracted from electronic patient record platforms; specifically, EMIS® and Vision®. These platforms are used to record information about patients including content of patient consultations, signs and symptoms, diagnoses, tests, medication prescriptions and referrals.”	Methods, ‘Study design, data sources, and participants’, page 6.
8	Authors wrote: “Patients with less than 3 months of follow-up time in the CPRD were excluded from our analyses”. This point should be discussed as a potential bias because most severe patients could have been lost.	In terms of loss of follow-up among patients with serious adverse outcomes, the risk of suicide within 3 months of an index self-harm episode, while higher than in the general population, is a rare outcome in absolute terms. While we were not able to study suicide outcomes in the present study, existing evidence suggests the cumulative incidence of suicide, within a year of a self-harm episode recorded in primary care, is around 0.5%.² Furthermore, there is no evidence that this proportion would differ between the COVID-19 study period and the pre-COVID-19 comparison group. We have added the following limitation to our Discussion: “Finally, we were unable to examine suicide deaths and other causes of mortality in this study due to	10

		unavailability of linked mortality records at the time of analysis.”	
9	Exposures, outcomes and covariates should be described in detail.	We have added the following details to the ‘Exposures, outcomes and covariates’ section: “Codes relating to intentional self-poisoning and self-injury episodes, of varying degrees of suicidal intent, were included”... “We examined any psychotropic medication and specific psychotropic medication types including antidepressants, antipsychotics, anxiolytic/hypnotics, mood stabilisers and stimulants.” We have also provided links to code lists to describe how exposures and outcomes were defined and classified.	7
10	Read, SNOMED and EMIS codes should be briefly explained. The same for FHSA and NHS fields.	We have added the following description of the clinical coding systems to the ‘Exposures, outcomes and covariates’ subsection of the Methods: “SNOMED CT ³ is a clinical vocabulary that is readable by computers. Used internationally, it is the recommended structured clinical vocabulary to record electronic patient information in the National Health Service in England. EMIS and Read codes are further coding systems used to capture clinical terms used in patient records. ⁴” The FHSA and NHS fields are simply variables available in CPRD data; we have clarified this in the manuscript (‘Exposures, outcomes and covariates’ subsection, Methods): “Information on referral to mental health services were identified using two CPRD fields: a ‘psychiatry’ code in the Family Health Services Authority (FHSA) speciality variable and codes of ‘mental illness’, ‘child	7

		and adolescent psychiatry', 'forensic psychiatry', 'psychotherapy', 'old age psychiatry', 'clinical psychology', 'adult psychiatry' and 'community psychiatric nurse' in the National Health Service (NHS) specialty field."	
11	Authors wrote: "A panel of four service users and carers with lived experience of health services following self-harm collaborated with the research team to plan the study and interpret results". May you explain a little bit more? Thanks.	We have added the following detail to the "Patient and public involvement" subsection of the Methods: "Panel members reviewed findings based on their experiences of health services for self-harm and the COVID-19 pandemic and its associated societal restrictions. Over two workshops, panel members met with the study's lead researcher (SS) to review and interpret its results, and provide feedback on their visual presentation. The group is linked with the National Institute Health Research Greater Manchester Patient Safety Translational Research Centre (NIHR GM PSTRC)."	8
12	Please add power analysis.	We did not conduct a power analysis for this study. When designing the study, we considered the feasibility of the study based on expected numbers of primary care-recorded self-harm episodes previously examined. In this instance, with an expected cohort size of 4000+ patients who have self-harmed during the COVID observation period, we knew that we would have abundant statistical power to examine these commonly occurring clinical management outcomes. Our original analyses of referral likelihood, conducted only in the GOLD dataset (as per the results presented in the submitted manuscript), were somewhat underpowered. We've rectified that deficiency now by conducting these analyses in the much larger pooled GOLD + Aurum dataset (see Table 2, 'Additional revisions').	7

		We preserved confidentiality by not disclosing cell counts less than 10. We have added the following detail to 'Exposures, outcomes and covariates': “Subgroup categories were derived to avoid reporting cell counts less than 10; if cell counts were found to be less than 10, subgroups were collapsed.”	
13	IMD should be better described.	The Index of Multiple Deprivation is a relative measure of area-level deprivation. The postcode of each general practice was linked to the IMD score of its corresponding lower super output area in England & Wales (approximate median population size: 1500 residents), super output area in Northern Ireland (approximate median population size: 2100 residents) or datazone in Scotland (containing a population of between 500 and 1000). The Index of Multiple Deprivation is a single score derived from seven domains of deprivation measures: income, employment, education, health, crime, barriers to housing and services, and living environment. We have added the following description to the 'Study design, data sources, and participants' section: “The Index of Multiple Deprivation is a single score derived from seven domains of area-based deprivation measures: income, employment, education, health, crime, barriers to housing and services, and living environment. The IMD is a relative measure of deprivation between areas. The postcode of the general practice was linked to the IMD score of its corresponding lower super output area in England and Wales (an	6

	Reviewer 2		
17	The manuscript presents an interesting and well-described population-based cohort study in primary care about clinical management following self-harm during the first wave of Covid-19 in UK. Results are important to describe that clinical management and referral to mental health service in that particular period. It is interesting to note that the management of self-harm was broadly similar to the management in the pre-pandemic period, with less referral to mental health services.	Thank you for this positive feedback on our study.	
18	- The management of self-harm includes a comprehensive mental assessment but also a management of the clinical situation associated with self-harm. I am embarrassed that care seems to be limited to the prescription of psychotropic drugs. GPs consultations and practice nurse consultations are the main component. I suggest changing the order of the objectives (and in the presentation of results, in tables) and putting (iii) as the	In our study we have utilised the complete coded information that was available to us in the electronic health records. As well as prescribing medication to patients and referring a small subset of them onto mental health services, GPs and practice nurses no doubt provide lots of effective care and support to their patients who have harmed themselves. Much of this information would be systematically coded though. For instance, we cannot examine how frequently these patients have seen a practice-based therapist or counsellor, as this information is known to be poorly captured in the CPRD records. Our ordering of the results was not intended to indicate the ranking of their priority as self-harm management. However, we are happy	Tables 2-4 and Results, pages 8-9.

	second objective and (ii) as the third....	to change the order of our reporting, which we have done.	
19	- Is it possible to know how many GPs consultations and practice nurse consultations was realized?	If the reviewer is referring to consultations that the patient did not attend, then we do not know if someone failed to attend a scheduled consultation unless explicitly recorded in their patient record (which isn't routinely or reliably done). Therefore, it is not possible to report this.	
20	- Is it possible to present data about the other diagnosis (than self-harm) established by GPs? About suicidal intent?	Self-harm is a damaging and risky a behaviour that is strongly linked with poor mental health, but it is not a diagnosable illness. Code lists for mental illness diagnoses such as depression and anxiety disorders, personality disorders, and eating disorders have been developed for use with the CPRD, but this was not the focus of this particular study. The information that is captured routinely in electronic health records does not include standardised measurement of suicidal intent for patients who have harmed themselves. The broad range of codes that we applied to identify self-harming behaviour will have encompassed a broad spectrum of severity entailing greatly varying degrees of suicidal intent. However, suicidal intent specific to each self-harm episode could not be examined in this study. We have added the following limitation to the Discussion: “Suicidal intent specific to each self-harm episode could not be examined in this study.”	10
21	- Results about psychotropic medication type (almost antidepressant) should be presented not only in supplementary	We have moved this table to the main manuscript (now Table 5).	17

	material. They deserve to be presented in the abstract, the results and in the discussion (comparison with others studies).		
22	- It would be interesting to describe further the four service users and careers' involvement in the study	Please see our response to point 11, above.	

Table 2: Additional revisions

(i) The initially submitted manuscript reported results pertaining to probability of referral to mental health services for general practices in Northern Ireland, Scotland and Wales (but not England), with these analyses carried out only in the CPRD GOLD dataset. Since submission of the manuscript, algorithms for running these analyses in the much larger Aurum + GOLD pooled dataset have been developed. In the revised manuscript, we also report results regarding probability of referral to mental health services from this pooled dataset, which now includes information from general practices in England (as well as Northern Ireland, Scotland and Wales).

(ii) Since presenting our findings at closed seminars with other experts in the field, we have received feedback on our results relating to likelihood of remote consultation by deprivation level. We now discuss these findings in more detail.

Addition	Specific changes made	Page no. in revised paper
(i) Reporting of probability of referral to mental health services using the larger pooled CPRD Aurum + GOLD dataset, which now includes information from practices in England as well as Northern Ireland, Scotland and Wales.	We have amended Table 3 and the Results section to reflect new analyses conducted in the pooled Aurum + GOLD datasets regarding probability of referral to mental health services: "Overall, 4.2% of patients (179/4238) were referred to mental health services in the COVID-19 first-wave cohort, a significant reduction versus the probability observed in the pre-pandemic comparison cohort (6.1%; ratio 0.70, CI 0.60, 0.81) (Table 3). The reduction in likelihood of being referred to mental health services was not observed for patients aged 65 years and over (ratio 1.66, CI 0.98 to 2.80), p-value for effect modification by age group = 0.01."	8-9 and Table 3

	We have also amended the text that pertains to these results in the Abstract: “The proportion of patients referred to mental health services in the COVID-19 cohort (4.2%) was around two-thirds of that in the pre-pandemic cohort (6.1%).”	2
(ii) Additional detail on likelihood of remote consultation by practice-level deprivation.	Abstract: “We found no evidence that the pandemic was associated with widening digital exclusion in terms of access to remote consultations.” Results: “With respect to remote consultation specifically, there was no difference between practice deprivation-level and likelihood of receiving this management.” Discussion: “Similarly, while there was a clear deprivation gradient in likelihood of remote consultation for self-harm prior to the pandemic, with those in areas of lower deprivation less likely to have remote consultation, the abrupt switch to remote consultations once the COVID-19 crisis had commenced did not lead to widening of existing inequalities in this respect. Evidence conducted prior to the COVID-19 pandemic indicated there was potential for the growth in virtual consultations to widen existing disparities in access to health. (24) Our findings suggest that for people who did seek help, there was no exacerbation of existing inequalities due to COVID-19 in these particular aspects of primary care clinical management, despite the growth in remote consultation numbers. In other words, we found no evidence of digital exclusion from self-harm care during the pandemic due to socioeconomic influences.” Conclusions: “We found no evidence of digital exclusion due to socioeconomic influences during the pandemic, in terms of likelihood of remote consultation.”	3 8 10 11

References

1. Benchimol EI, Smeeth L, Guttman A, et al. The REporting of studies Conducted using Observational Routinely-collected health Data (RECORD) Statement. Plos Medicine 2015; 12(10).
2. Carr MJ, Ashcroft DM, Kontopantelis E, et al. Premature Death Among Primary Care Patients With a History of Self-Harm. Annals of Family Medicine 2017; 15(3): 246-54.
3. NHS Digital. SNOMED CT. 2021. <https://digital.nhs.uk/services/terminology-and-classifications/snomed-ct> (accessed 02.11.2021).
4. NHS Digital. UK Read Code. 2015. <https://data.gov.uk/dataset/f262aa32-9c4e-44f1-99eb-4900deada7a4/uk-read-code>.

VERSION 2 – REVIEW

REVIEWER	Younès, Nadia EA 40-47 University of Versailles Saint-Quentin
REVIEW RETURNED	17-Dec-2021
GENERAL COMMENTS	Thanks for the revisions